observational astronomy

light pollution, astronomical observatories, site assessment, spatial planning, lighting, environmental protection

**Author for correspondence:**
Salvador Bará
e-mail: salva.bara@usc.gal

# A linear systems approach to protect the night sky: implications for current and future regulations

Fabio Falchi[1,2] and Salvador Bará[1]

[1]Dept. de Física Aplicada, Universidade de Santiago de Compostela, 15782 Santiago de Compostela, Galicia, Spain
[2]Istituto di Scienza e Tecnologia dell'Inquinamento Luminoso (Light Pollution Science and Technology Institute), 36016 Thiene, Italy

 SB, 0000-0003-1274-8043

The persistent increase of artificial light emissions is causing a progressive brightening of the night sky in most regions of the world. This process is a threat for the long-term sustainability of the scientific and educational activity of ground-based astronomical observatories operating in the optical range. Huge investments in building, scientific and technical workforce, equipment and maintenance can be at risk if the increasing light pollution levels hinder the capability of carrying out the top-level scientific observations for which these key scientific infrastructures were built. Light pollution has other negative consequences, as e.g. biodiversity endangering and the loss of the starry sky for recreational, touristic and preservation of cultural heritage. The traditional light pollution mitigation approach is based on imposing conditions on the photometry of individual sources, but the aggregated effects of all sources in the territory surrounding the observatories are seldom addressed in the regulations. We propose that this approach shall be complemented with a top-down, ambient artificial skyglow immission limits strategy, whereby clear limits are established to the admissible deterioration of the night sky above the observatories. We describe the general form of the indicators that can be employed to this end, and develop linear models relating their values to the artificial emissions across the territory. This approach can be easily applied to other protection needs, like e.g. to protect nocturnal ecosystems, and it is expected to be useful for making informed decisions on public lighting, in the context of wider spatial planning projects.

# 1. Introduction

The persistent increase in the emissions of artificial light [1] is giving rise to a progressive deterioration of the natural night,

with detrimental consequences for the environment [2,3], science [4], cultural heritage [5], energy consumption [6,7], and, arguably, human health [8,9], actively studied within the interdisciplinary field of light pollution research [10]. One of the most visible effects of light pollution is the loss of the darkness of the night sky, due to the atmospheric scattering of the light emitted by artificial sources [11–14]. In the visible band, this scattered light reduces the luminance contrast between the celestial objects and their immediate surroundings, preventing the vision of faint objects that, in the absence of artificial light, would be clearly detected. The consequences for the preservation of the intangible cultural heritage associated with the contemplation of the starry skies are a matter of widespread concern [5]. The same effect of loss of contrast takes place in any other photometric band affected by the visible and near-infrared emissions from outdoor lighting sources.

Light pollution poses a serious threat for the long-term sustainability of first-class astronomical observatories. These key scientific infrastructures, in which huge investments have been and are made throughout decades, may find their work jeopardized by the increased levels of scattered light from urban populations, rural settings, industrial areas, mines, and the road and transportation networks in their area of influence. The effects of light pollution can be detected at hundreds of kilometres from the sources. Any increase of the artificial light emissions cooperates to hinder the possibility of carrying out the top-level science for which the observatories were built and equipped. This phenomenon is not new: the first Western modern observatories, built around the eighteenth century AD were located within the main towns or their immediate surroundings. The sky at these locations brightened along the nineteenth and, particularly, the twentieth centuries, making it necessary to relocate the observatories in remote sites less affected by light pollution (and with lower turbulence and higher transparency than urban skies can provide). The history of the great Californian observatories is paradigmatic. Lick, Mount Wilson and Palomar mountain observatories started with nearly pristine skies when their construction began, around 1880, 1905 and 1936, respectively. They experienced an increase of pollution that brightened their skies to about three, eight and two times compared to the natural values. The process of deterioration of the night sky did not stop in the twentieth century. Nowadays, light from artificial sources can be detected, with different degrees of intensity, in practically all ground-based observatories. As with many other environmental threats, the persistent increase of this one may easily get unnoticed in short periods of time to the casual observer, but its accumulated effects are clearly noticeable in periods of the order of several years. Very often, when these effects are finally evident, the situation gets considerably difficult to remediate.

The rising awareness about this problem led several governments to issue a set of legal regulations aiming to curb the growth of light pollution, particularly in the vicinity of the observatories. Some examples of it are the Spanish Law 31/1998 'about the protection of the astronomical quality of the Observatories of the Institute of Astrophysics of Canary Islands', and the recent norms issued by the government of Chile. Additional regulations have been approved by several countries to address the environmental and energy expenditure aspects of light pollution (Italian regions, Slovenia etc.). Most of the existing regulations, however, are almost exclusively based in what can be termed the 'individual source' approach, that is, they set relatively strict limits to the amount of light, directionality, spectral composition and switching times of the individual light sources (either individual luminaires, or individual installations comprising generally of a small set of luminaires), in an effort to contain the total emissions of light. Slovenia added to these rules a cap in *per capita* energy consumption but note that this does not stop the possible increase of emitted light, as light efficiency increases over time with technology advancements. Two important improvements to this general trend shall be mentioned. The 1989 lighting code of Flagstaff (AZ, USA), established global limits on the amount of lamp lumen per acre in four zones near the US Naval Observatory Flagstaff Station and Lowell Observatory's Anderson Mesa, to limit the brightening of the sky over these sites [15]. Similarly, the 2012 City of Tucson/Pima County Outdoor Lighting Code (AZ, USA), establishes a lumen cap on the total outdoor light output per acre of developed areas (https://webcms.pima.gov/UserFiles/Servers/Server_6/File/Government/Development%20Services/Building/OLC.pdf) 'to preserve the relationship of the residents of the City of Tucson, Arizona and Pima County, Arizona to their unique desert environment through protection of access to the dark night sky'.

It seems clear that imposing conditions on individual sources or small patches of the territory as the only strategy does not guarantee by itself the preservation of the quality of the skies, unless the total emissions are kept below the critical level beyond which the damage to the observatories' skies would be noticeable. In this paper, we develop a complementary, top-to-bottom approach, aimed to effectively ensure that the light pollution levels of the observatory skies do not surpass some agreed critical limiting values.

This approach is based on the use of quantitative indicators of the quality of the night sky, continuous night sky brightness monitoring to assess compliance, and decision-making support tools to determine the maximum amount of light emissions in the surrounding territories that are compatible with the preservation of the observatory skies. The practical application of this approach requires addressing three main issues: (i) choosing the appropriate indicators of the quality of the night sky, and establishing their acceptable limiting values, (ii) developing a quantitative model relating the values of these indicators to the artificial light sources existing in the region around the observatory, and (iii) allocating among the surrounding municipalities and other local administrative bodies the possible quota of new light emissions (in case the present value of the indicators still did not reach the limiting values, and it could be accepted to increase the emissions in spite of its detrimental consequences) or distributing the burden of reducing the emissions (in case the critical values of the indicators have been surpassed).

The best light pollution indicators for astronomical observatories are still under discussion. A definite proposal was made some time ago by the International Astronomical Union (IAU) [16]. This provision establishes that 'the increase in sky brightness at 45° elevation due to artificial light scattered from clear sky should not exceed 10% of the lowest natural level in any part of the spectrum between wavelengths 300 and 1000 nm except for the spectral line emission from low-pressure sodium lamps as set out in Recommendation 2 (…)'. Although this provision would very likely need to be updated for the current lighting technologies, it is a good example of the kind of global indicators to which we refer in this work.

In this paper (§2), we focus on developing the basic aspects of the point (ii) above, that is, the model linking the values of arbitrary linear indicators to the emissions of the sources. Some general considerations on admissible deterioration limits, quota allocation and long-term planning are made in §3. Several results used in §2 are formalized in the appendix.

# 2. The linear propagation of light pollution

## 2.1. Sky quality indicators and the point spread function of artificial lights

Ground-based astrophysical observations are carried out with detectors operating in several photometric bands [17], with different fields of view and angular resolutions depending on the instrument type, and with different scientific goals. However, they share something in common: the basic physical quantity at the root of the signal provided by most detectors in the optical frequency range is the total spectral radiance $L_T(\mathbf{r}'', \boldsymbol{\alpha}'', \lambda)$, with units $W\,m^{-2}\,sr^{-1}\,nm^{-1}$, or photon $s^{-1}\,m^{-2}\,sr^{-1}\,nm^{-1}$. The radiance field provides information about the radiant power (W or photon $s^{-1}$) incident at point $\mathbf{r}''$ from the direction $\boldsymbol{\alpha}'' = (z'', \varphi'')$, being $z''$ the zenith angle and $\varphi''$ the azimuth, per unit surface ($m^2$) around $\mathbf{r}''$, per unit solid angle (sr) around $\boldsymbol{\alpha}''$ and per unit wavelength interval (nm) around $\lambda$. Additional variables of $L_T$ like the time, $t$, and the polarization state of the light, $\mathbf{p}$, are not explicitly included here for the sake of simplicity. Most raw measurements provided by the detectors can be expressed in terms of this basic physical function.

In what follows we use some results whose formalization can be found in the appendix. Additional information on the physical quantities and notation used here can be found in references [18,19]. Due to their finite aperture, field of view and spectral resolution, the raw signal $B_T$ provided by any detector, working within its linearity range or after the nonlinearities have been compensated for, is proportional to a spatial, angular and spectral-weighted average of the incident radiance. This can be expressed as $B_T = \mathcal{L}_1\{L_T\}$, where $\mathcal{L}_1$ is a linear operator whose particular form depends on the photometric quantity being measured and the particular specifications of the instrument. $B_T$ can be any quantity linearly related to the radiance, for instance the brightness of the night sky in any particular observation direction (e.g. the zenith) and spectral band, the average brightness of the upper hemisphere or of some region around the horizon, the horizontal irradiance, the vertical or averaged vertical irradiance and many others, expressed either in absolute values or in values relative to some pre-defined reference level.

The total radiance $L_T$ incident on the observing instrument is itself composed of two terms: the natural radiance, $L_N$, which carries information about the universe around us, and the anthropogenic one, $L$, the artificial radiance produced by the atmospheric scattering of the light from outdoor light sources. The detected signal is then given by $B_T = \mathcal{L}_1\{L_N\} + \mathcal{L}_1\{L\} \equiv B_N + B$, where $B_N$ is the component carrying science information about the natural sources (comprising celestial bodies,

zodiacal light and airglow) [20,21] and $B$ is the component associated with light pollution. To fully exploit the performance of the available instruments, minimizing the value of $B$ is a must. The Michelson contrast between the light coming from the direction of the science object and its immediate surroundings is given by $\gamma = 1/[1 + (2B/B_N)]$, being thus reduced, sometimes considerably, with respect to its ideal value of unity. Similarly, the Weber contrast $w = B_N/B$, widely used in vision science, decreases as the light pollution component increases. Even if the value of $B$ could be precisely known for the site and the moment of the observations, based on theoretical models, and then subtracted from $B_T$ to obtain $B_N$, its presence wastes part of the dynamical range of the detector due to the combined effects of quantization and photon noise, reducing its effective resolution in a way that cannot be fully compensated by modifying the aperture settings or exposure times. This also applies to naked-eye observations of the starry sky, since the changes in the eye sensitivity afforded by its luminance adaptation capability (from photopic to scotopic, across the mesopic range [22]) cannot compensate for this effect, even if some marginal increase or decrease of the luminance contrast can be achieved as a consequence of the spectral sensitivity shift associated with the different luminance adaptation levels.

The values of $B$, either absolute or relative to $B_N$, are thus suitable indicators of the quality of the skies above observatories, in what regards the detrimental effects of light pollution. The particular form of the optimum indicator or set of indicators for a given observatory is contingent on the quality criteria adopted in each case by the observatory managers, taking into account the specifications of the operating instruments and the characteristics of the celestial bodies to be studied with them. Irrespective of their detailed form, however, they can be expressed as the action of the linear operator $\mathcal{L}_1$ on the artificial radiance $L$, such that $B = \mathcal{L}_1\{L\}$. A few exceptions exist to this rule: some potentially useful indicators, as e.g. the maximum artificial radiance of the night sky or its average brightness expressed in mag arcsec$^{-2}$ [22], cannot be obtained linearly from $L$ since neither the max function nor the logarithmic scale in mag arcsec$^{-2}$ are linear functions. However, these particular indicators can be straightforwardly computed once the values of $L$ have been determined using a linear transformation on the sources' radiance (see appendix).

As shown in equation (A 4), the artificial radiance at the observer location, $L$, is linearly dependent on the radiance $L_s$ of the artificial light sources located in its surroundings, such that we can write $L = \mathcal{L}_2\{L_s\}$. Combining this equation with the one for the light pollution indicator, $B = \mathcal{L}_1\{L\}$, one immediately gets $B = \mathcal{L}_1\{\mathcal{L}_2\{L_s\}\} = \mathcal{L}\{L_s\}$, where $\mathcal{L} = \mathcal{L}_1 \circ \mathcal{L}_2$ is the linear operator resulting from the composition of the actions of $\mathcal{L}_1$ and $\mathcal{L}_2$. The general form of the $\mathcal{L}$ operator is shown in the appendix. As it is also shown there, under the very general assumption that the sources are factorable, that is, that their radiance can be expressed as the product of a spatial and a spectral-angular term such that $L_s(\mathbf{r}', \boldsymbol{\alpha}', \lambda) = L_1(\mathbf{r}')\,L_2(\boldsymbol{\alpha}', \lambda)$, the value of the indicator $B(\mathbf{r})$ at an observatory located at $\mathbf{r}$ is related to the spatial distribution of the radiance $L_1(\mathbf{r}')$ of the artificial light sources by

$$B(\mathbf{r}) = \int_{A'} K(\mathbf{r},\mathbf{r}')\,L_1(\mathbf{r}')\,\mathrm{d}^2\mathbf{r}', \tag{2.1}$$

where $K(\mathbf{r},\mathbf{r}')$ is the point spread function (PSF) of the light pollution produced by the artificial light sources, that is, the amount in which a unit radiance light source located at $\mathbf{r}'$ contributes to the total value of $B$ at $\mathbf{r}$ [18,19]. The PSF depends on the choice of the indicator used to describe the night sky quality, the specifications of the detector, the atmospheric conditions, in particular, the distribution and type of aerosols, the ground spectral reflectance, the presence or not of obstacles either natural (relief, trees etc.) or artificial (buildings) and the angular and spectral radiating patterns of the luminaires, $L_2(\boldsymbol{\alpha}', \lambda)$. The integral is extended to $A'$, the region around the observatory encompassing all relevant artificial sources, being $\mathrm{d}^2\mathbf{r}'$ the surface element of the territory.

Whereas the exact value of the PSF (for a given set of atmospheric conditions) may be strongly dependent on the individual values of $\mathbf{r}$ and $\mathbf{r}'$, in many cases of practical interest the PSF can be approximately considered to be shift-invariant, that is, $K(\mathbf{r},\mathbf{r}') = K(\mathbf{r} - \mathbf{r}')$ so that it only depends on the relative position of the observatory with respect to each source. This approximation can be applied, e.g. to a homogeneous territory with a layered atmosphere, or if an averaged PSF is computed for a given set of atmospheric and geographical conditions. If the PSF is shift-invariant the superposition integral in equation (2.1) becomes a two-dimensional convolution and the full toolbox of Fourier transform methods can be applied to efficiently calculate the value of the indicator $B(\mathbf{r})$ over a wide area of the territory in a single step [19], instead of computing it sequentially for each observing point by a repeated application of equation (2.1). This option is particularly useful if the value of the

indicator shall be calculated for all pixels of a large region, like a dark sky reserve, a national park, or a whole country or set of countries.

The $K(\mathbf{r},\mathbf{r}')$ PSF can be calculated by computing the effects of a single point source using suitable radiative transfer models and appropriate atmospheric characterization [23]. Several PSFs have been developed in the literature for the zenithal brightness, the brightness in arbitrary directions and the average brightness of the upper hemisphere relative to its nominal natural value [24–31]. The same procedures can be applied to determine the PSF for other linear indicators of the quality of the night sky.

## 2.2. Changes in the sky quality indicators due to changes in lighting installations

The linear relationship in equation (2.1), suggests an easy way to assess the effects of changes in lighting installations. If the new installed luminaires have the same angular and spectral patterns as the existing ones, the PSF $K(\mathbf{r},\mathbf{r}')$, which implicitly depends on $L_2(\alpha', \lambda)$, will be the same for the old and new sources, and any incremental change $\Delta L_1(\mathbf{r}')$ in the emissions of the surrounding territory will give rise to a change in the indicator

$$\Delta B(\mathbf{r}) = \int_{A'} K(\mathbf{r},\mathbf{r}')\, \Delta L_1(\mathbf{r}')\, \mathrm{d}^2\mathbf{r}'. \tag{2.2}$$

Note that the condition of the equality of the angular and spectral radiation patterns of the new and the old installations is not required to be fulfilled by each individual light source, but by the aggregated emissions of all luminaires (including the reflections in pavements and façades) contained within the territory element $\mathrm{d}^2\mathbf{r}'$. In case these aggregated emissions would result in a substantially different form of the function $L_2(\alpha', \lambda)$, then it may be convenient to recalculate the form of the PSF $K(\mathbf{r},\mathbf{r}')$ that, as stated above and shown in equation (A 10) of the appendix, implicitly depends on it.

Note also that any local change in the lighting installations, $\Delta L_1(\mathbf{r}')$, univocally determines, via equation (2.2), the change that will experience the indicator, $\Delta B(\mathbf{r})$. However, the inverse is generally not true: any given change of the indicator can be achieved in multiple ways, by redistributing the artificial emissions among all elements of the territory. The basic reason is that the transformation in equation (2.2) cannot be inverted to get a unique $\Delta L_1(\mathbf{r}')$ from $\Delta B(\mathbf{r})$ if $\Delta B(\mathbf{r})$ is specified only for an individual observatory, $\mathbf{r}$. Infinite choices for $\Delta L_1(\mathbf{r}')$ are in that case possible, providing the same $\Delta B(\mathbf{r})$. This will be relevant for adopting decisions on how to allocate between different local communities the admissible increase of emissions or the required reductions thereof, something we address in more detail in §3.

## 2.3. Calculating the indicators with geographical information system

Equation (2.1) lends itself well to be interpreted in terms of georeferenced matrices, which can be processed with geographical information system software (GIS). Here, we recall the main steps, described in detail in [18,19]. The emissions function $L_1(\mathbf{r}')$ is usually available as a georeferenced raster with pixels of finite size, specified in some coordinate reference system [32]. Frequently these input files are provided in a WGS84 reference frame, with pixels of uniform latitude-longitude angular size (e.g. $15 \times 15$ arcsec$^2$, for the VIIRS-DNB composites [33]). In such cases it is convenient to reproject the files to a grid of pixels of uniform size in linear dimensions (m or km), like e.g. any of the universal transverse Mercator (UTM) projections used in the official cartography of many countries [32]. The integrand in equation (2.1) can be interpreted then as the pixel-wise product of an emissions matrix $\mathbf{L}$, an example of which is shown in figure 1$a$, whose elements are given by $L_{ij} = L_1(\mathbf{r}_{ij}')\, \sigma$, where $\sigma$ is the area (m$^2$) of the uniformly sized reprojected pixels, and a PSF matrix $\mathbf{K}$, figure 1$b$, whose elements are $K_{ij} = K(\mathbf{r},\mathbf{r}_{ij}')$, that is, the PSF function centred on the point of observation, $\mathbf{r}$, with coordinates reflected in the origin, $(x,y) \rightarrow (-x, -y)$. The element-wise product ("$\cdot$") of these matrices gives rise to the weighted sources matrix $\mathbf{W} = \mathbf{K} \cdot \mathbf{L}$, figure 1$c$, where each pixel $W_{ij} = K_{ij}L_{ij}$ displays the absolute contribution of its sources to the value of the overall indicator $B(\mathbf{r})$. The final value of $B(\mathbf{r})$ is given by the sum over all pixels of $\mathbf{W}$,

$$B(\mathbf{r}) = \sum_{i,j} K_{ij}L_{ij} = \sum_{i,j} W_{ij}. \tag{2.3}$$

Accordingly, any change in the amount of emissions of an individual pixel, $\Delta L_{ij}$, translates into a proportional change in the indicator, $[\Delta B(\mathbf{r})]_{ij} = K_{ij}\Delta L_{ij}$. Equal changes in the absolute emissions of different pixels will give rise to different changes in the indicator due to the different weighting factors $K_{ij}$.

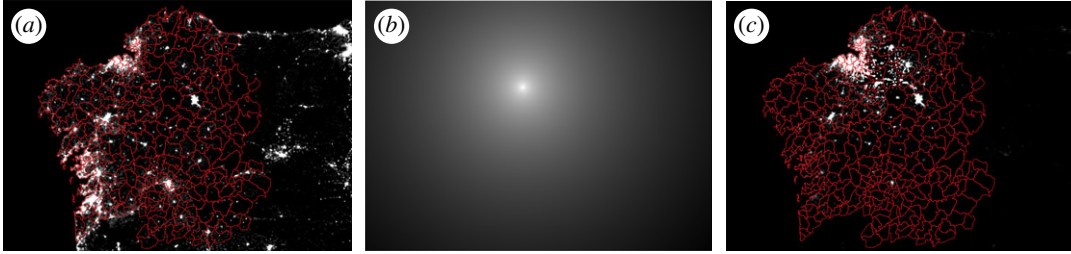

**Figure 1.** (*a*) The emissions matrix **L**, according to the VIIRS-DNB stable light sources [33], 2015 yearly composite, with the limits of the Galician municipalities superimposed on it (red). Each pixel of this matrix gives the radiance (nW cm$^{-2}$ sr$^{-1}$) detected by the VIIRS-DNB radiometer. (*b*) the PSF matrix **K** calculated for the artificial zenith sky brightness according to the model of Cinzano & Falchi [25] for an atmosphere with clarity index 1 (visibility 26 km) centred in the astronomical observation site of San Xoán, Guitiriz (43°13′31.17″ N, 7°55′37.44″ W), displayed here in a grey-level logarithmic scale. (*c*) The weighted sources matrix **W** = **K**·**L**. Each pixel of this matrix gives the absolute contribution of the light sources contained within it to the final value of the zenith sky brightness in the Johnson V band at the observing site, in radiance units.

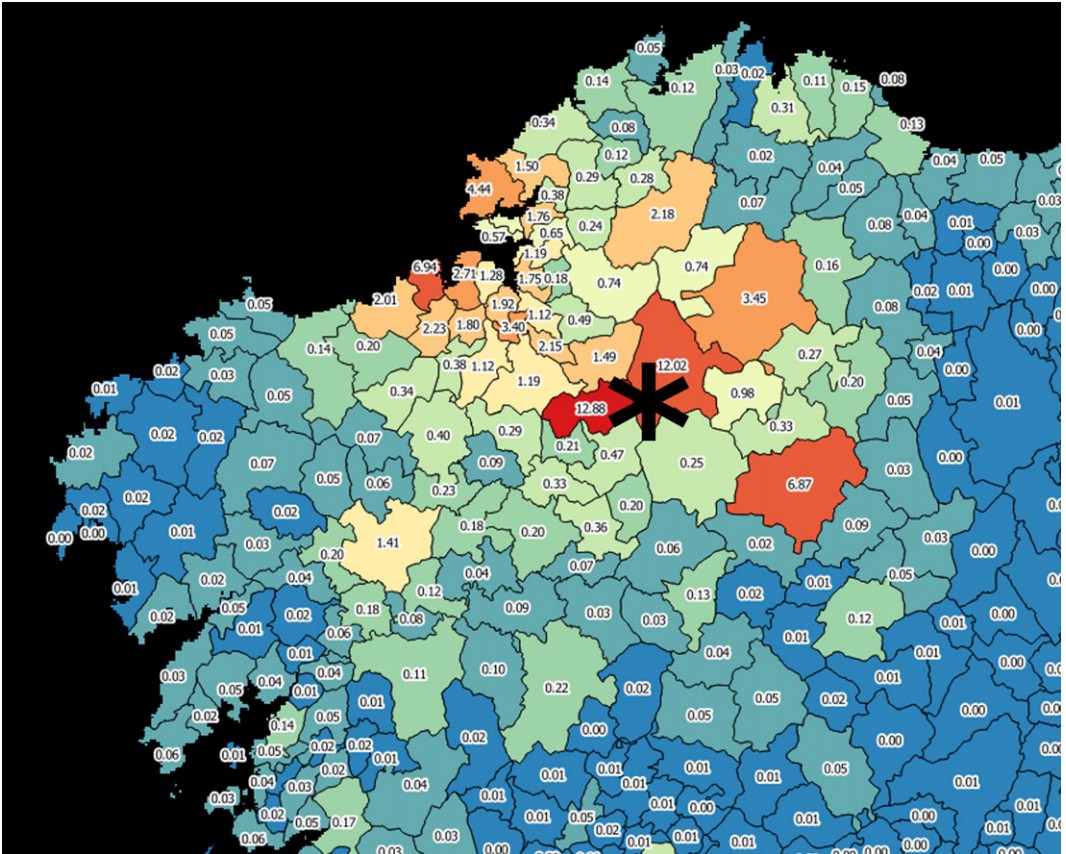

**Figure 2.** The relative contribution of the emissions of each municipality to the artificial zenith sky brightness at the Guitiriz observing site (asterisk).

For making informed decisions on public lighting it is often convenient to group the pixel contributions according to the administrative division of the territory, particularly in terms of the local bodies in charge of that service. Figure 1*a*,*c* displays in red the limits of the municipalities, which are the main responsible bodies of outdoor lighting in Galicia, an autonomous community within Spain. The absolute contributions of the individual pixels to the overall indicator for which they were calculated (the artificial zenith night sky brightness at an astronomical observations site in the municipality of Guitiriz, computed for the photometric Johnson V band in agreement with the Cinzano & Falchi model in [25]) can be added at the municipality level, resulting in the relative contribution map shown in figure 2, where the per cent contribution of the aggregated emissions of

each municipality is shown. These calculations, as well as the coordinate reference system reprojections that may be needed, can straightforwardly be made using free GIS applications, as e.g. QGIS [34]. The relative contribution of each municipality to the total value of the indicator allows to easily determine the changes that will experience the indicator for any projected increase or decrease of the municipality emissions relative to its present levels.

Some caution shall be exercised when selecting and using satellite radiance datasets to perform these calculations. For instance, the panchromatic VIIRS-DNB band is not sensitive to wavelengths shorter than 500 nm, missing the blue peak of phosphor-coated LED sources, whereas it detects the NIR peaks of high-pressure sodium lamps. The radiance detected by satellites is also dependent on atmospheric conditions and ground albedo variability [35]. These and other potential variability factors must be judiciously taken into account. Note that the relative contributions of municipalities or districts to the light pollution over the observatories calculated following the approach described in this paper are relatively robust against the spectral passband of VIIRS-DNB, if the mix of lighting technologies used in every pixel of the territory can be considered to be approximately the same. Regarding the atmospheric and ground albedo variability of the detected radiance, this makes it more difficult assessing the actual time evolution of the source emissions. However, it could be considered as an advantage rather than a drawback: as a matter of fact, the emissions relevant for light pollution are not just the raw emissions of the luminaires, but the effective amount of light radiated from each pixel of the territory. In this sense, the fact that VIIRS-DNB datasets record this variability could be a useful asset rather than something to correct.

# 3. Admissible indicator limits, emission quota allocation and long-term planning

The sustained increase of the artificial light emissions suggests that a preservation strategy based on imposing conditions on the photometric specifications of the individual luminaires is not, *per se*, sufficient to ensure that the quality of the night skies over the reference astronomical observatories (or any other protected area, which may well include the whole territory of a country) will be maintained in the long term. An effective protection of the night skies requires additional provisions, which we summarize here in the following three subsections:

## 3.1. Admissible indicator limits

Any effective public policy on light pollution mitigation requires adopting a clear decision about the levels of deterioration of the night sky considered not admissible (the *red lines* not to be surpassed). The increase of artificial sky brightness progressively hinders the capability of the observatories to perform top-level science tasks, and for each type of astrophysical study there are levels of light pollution that would not allow carrying it out with the required accuracy and precision (due to the reduction in the dynamic range and effective resolution of the detectors caused by the combined effects of quantization and photon noise). The observatory managers should then decide the limiting values of the indicators that can be considered admissible. This is a kind of decision that bears many aspects in common with the ones that have to be taken in other environmental problems, as e.g. setting limits on the harmful concentrations of particulate matter in the atmosphere (PM2.5, PM10). Regarding light emissions, if the present values of the indicators are smaller than the limits, adopting a criterion of prudence would be strongly advisable, by keeping the overall weighted emissions in or below their present values as a sensible strategy to better ensure the protection of the night skies. New installations could be made at the expense of decreasing in an equivalent amount the emissions in other places of the region. Conversely, if the present values of the indicators surpass the admissible values, a definite and planned transition process must be designed and put in practice, involving overall reductions in the weighted source emissions, by reducing the absolute emissions and/or by redistributing them spatially across the territory. Examples of present-day distributions of territorial contributions, from pixel level to whole municipalities, can be found in [18,31]. Note also that astronomers' goals are not the only ones that are at stake: environmental and landscape preservation concerns may advise adopting indicator limits even more restrictive than those required to ensure the quality of astronomical observations. In such cases, the strictest limits should be agreed and enforced. Finally, the quality indicators do not have to be restricted to a single-point site. They can be defined for whole zones of the territory, as e.g. in (https://www.epa.gov/visibility/regional-haze-program;

https://www.darksky.org/our-work/lighting/public-policy/mlo/). The general formalism for defining such kind of territorially weighted linear indicators has been described in [18].

## 3.2. Emission quota allocation

As described in §2.2, any increment or reduction of the value of a light pollution indicator can be achieved in multiple ways, by readjusting the light emissions of the intervening elements of the territory. In any case, the territorial distribution of the increment or decrement of emissions must be made taking into account a wide variety of social factors. A given increase or reduction of emissions could be allocated uniformly for all elements of the territory, or in an amount proportional to the present emissions. However, these easy mathematical options would surely fail to fulfil basic criteria of social equity. The small communities located close to astronomical observatories, for instance, contribute to the deterioration of the night skies above the scientific installation in a proportionally bigger amount than others located far away, for the same absolute level of emissions, due to the larger values of their $K_{ij}$ factors. Should these communities bear the charge of reducing their emissions in the same proportion as others, or should they be allowed to make lesser reductions (or even moderate increments where needed) at the expense of larger reductions in distant communities?

## 3.3. Long-term planning

Long-term planning is a key issue. In many cases, any individual operation of remodelling of outdoor lighting will imply a modest increase of the emissions ($\Delta L_{ij}$) in comparison with their global present values ($L_{ij}$). The change in the overall light pollution indicators after such remodelling operation is then expected to be relatively small. However, any increase of an indicator, albeit modest, spends part of the available light emissions increase budget. Any new installation authorized nowadays will effectively reduce the available quota of admissible additional emissions, hindering the margins for authorizing new projects in the future. The public bodies in charge of outdoor lighting shall then have a strategic plan of development (either allowing increasing emissions or ensuring to reduce them, depending on the cases) that contemplates the long-term evolution of both the science tasks and the social needs. This is a dimension that shall be included in any spatial planning project in the concerned territories. Several cap and trade mechanisms can be used to adaptively correct the initial quota allocation in case of changing priorities or emerging social needs.

These three requirements are science-informed ones, but they are essentially political issues. They reflect the tension between conflicting interests and needs [36–41] and should be addressed with the democratic participation of stakeholders, as a relevant aspect of community management and spatial planning.

# 4. Conclusion

The preservation of the darkness of the natural skies is a requisite for the long-term sustainability of the operations of astronomical observatories. The increasing amount of light pollution represents a threat that shall be addressed before the situation deteriorates beyond permissible levels. This requires defining a suitable set of sky quality indicators and agreeing their admissible limits. The indicator values can be related to the radiance of the artificial light sources through linear models that allow to assess the absolute and relative contribution of each patch of the territory to the deterioration of the night sky above the observatory. Since the relationship between artificial sky brightness and light emissions is linear, any increase of the emissions contributes to further deteriorating the present levels of light pollution. Simply putting limits on the characteristics of individual lighting installations, albeit necessary, is insufficient to address this problem. Setting overall indicator limits, distributing the emissions quota among the affected communities, and planning with a long-term perspective are necessary requisites of an effective light pollution control policy.

Data accessibility. All data sources used in this work are in the public domain. No additional materials would be required to conduct an attempt at replication. VIIRS-DNB composites are available, among other options, from [33]. The Galician municipalities' border shapefile can be downloaded from http://centrodedescargas.cnig.es/CentroDescargas/equipamiento.do
Authors' contributions. Both authors contributed equally to this work.
Competing interests. We have no competing interests.

Funding. S.B. acknowledges support from Xunta de Galicia grant no. ED431B 2020/29.

Acknowledgements. We wish to express our thanks to the reviewers of this paper for their useful comments and suggestions.

# Appendix A

## A.1. Spectral radiance and night sky brightness measurements

Due to their finite spatial, angular and spectral resolution, each independent channel of an astrophysical detector located at $\mathbf{r}$ and pointing towards $\boldsymbol{\alpha}$ provides a total signal $B_T(\mathbf{r},\boldsymbol{\alpha})$ that, within its linear response range and leaving aside noise, is proportional to a weighted average of the incident radiance, $L_T(\mathbf{r}'', \boldsymbol{\alpha}'', \lambda)$, which can be expressed as

$$B_T(\mathbf{r},\boldsymbol{\alpha}) = \int_\Lambda \int_A \int_\Omega R(\mathbf{r},\boldsymbol{\alpha}; \mathbf{r}''\boldsymbol{\alpha}''; \lambda) L_T(\mathbf{r}'',\boldsymbol{\alpha}'', \lambda)\, \mathrm{d}^2\boldsymbol{\alpha}'' \mathrm{d}^2\mathbf{r}'' \mathrm{d}\lambda, \qquad (A\,1)$$

where $R(\mathbf{r},\boldsymbol{\alpha}; \mathbf{r}'',\boldsymbol{\alpha}''; \lambda)$ is a combined spectral, aperture and field-of-view sensitivity function describing the relative response of the instrument to the radiance of wavelength $\lambda$ incident from the direction $\boldsymbol{\alpha}''$ on the point $\mathbf{r}''$ of the input pupil, when the instrument, located at $\mathbf{r}$, points towards $\boldsymbol{\alpha}$. The angular integral is extended to $\Omega$, the whole set of relevant directions, being $\mathrm{d}^2\boldsymbol{\alpha}'' = \sin z''\, \mathrm{d}z''\, \mathrm{d}\varphi''$ the solid angle element in a spherical coordinate system with its polar axis coincident with the pointing direction of the detector. The spatial integral is extended to the area $A$ of the input pupil of the system, being $\mathrm{d}^2\mathbf{r}'' = \mathrm{d}x'' \mathrm{d}y''$ the area element in Cartesian coordinates. The wavelength integral is extended to the whole spectrum, $\Lambda$.

Equation (A 1) is in fact a very general one, encompassing a wide variety of quantities measured by astrophysical instruments (see additional formalization details in [42]). The different types of instruments and their associated photometric bands are characterized by the $R$ function. In many cases of practical interest this function can be factored as the product of independent terms such that $R(\mathbf{r},\boldsymbol{\alpha}; \mathbf{r}'',\boldsymbol{\alpha}''; \lambda) = P(\mathbf{r},\mathbf{r}'')F(\boldsymbol{\alpha},\boldsymbol{\alpha}'')S(\lambda)$, where $P(\mathbf{r},\mathbf{r}'')$ is the input pupil function, $F(\boldsymbol{\alpha},\boldsymbol{\alpha}'')$ is the function characterizing the instrument field of view, including vignetting, and $S(\lambda)$ is the photometric band in which the observations are carried out. $P$ and $F$ are usually normalized such that their volumes are unity, whereas $S$ is traditionally normalized to 1 at its peak. For instance, for a point-like detector with a very small field of view that measures the night sky radiance in the Johnson–Cousins $V$ band, we have $P(\mathbf{r},\mathbf{r}'') = \delta(\mathbf{r} - \mathbf{r}'')$ and $F(\boldsymbol{\alpha},\boldsymbol{\alpha}'') = \delta(\boldsymbol{\alpha} - \boldsymbol{\alpha}'')$, where the $\delta$ stand for Dirac-delta distributions, and $S(\lambda) = V(\lambda)$, such that the brightness of the night sky measured at that location and in that direction is given by the usual expression

$$B_T(\mathbf{r},\boldsymbol{\alpha}) = \int_\Lambda V(\lambda) L_T(\mathbf{r},\boldsymbol{\alpha}, \lambda)\, \mathrm{d}\lambda, \qquad (A\,2)$$

where $B_T(\mathbf{r},\boldsymbol{\alpha})$, which natively has radiance units (W m$^{-2}$ sr$^{-1}$, or photon s$^{-1}$ m$^{-2}$ sr$^{-1}$), can be expressed in the traditional logarithmic scale of *magnitudes per square arcsecond* (mag arcsec$^{-2}$) by means of a straightforward transformation [22]. Note that many additional photometric quantities can be expressed as particular cases of equation (A 1). For instance, the irradiance $E_T(\mathbf{r},\boldsymbol{\alpha})$ within the $S(\lambda)$ band (units W m$^{-2}$, or photon s$^{-1}$ m$^{-2}$), produced on the input pupil of the instrument by a patch of the sky of solid angle $\omega$ can be written as

$$E_T(\mathbf{r},\boldsymbol{\alpha}) = \int_\Lambda S(\lambda) \int_\omega L_T(\mathbf{r},\boldsymbol{\alpha}'', \lambda) \cos(\boldsymbol{\alpha},\boldsymbol{\alpha}'')\mathrm{d}^2\boldsymbol{\alpha}''\, \mathrm{d}\lambda, \qquad (A\,3)$$

and the same holds for many photometric quantities routinely measured in astrophysics and light pollution research (for a description of some of the latter, see [43]).

A key feature of equation (A 1) is that the detected signal $B_T(\mathbf{r},\boldsymbol{\alpha})$ can be expressed as the result of the action of an integral linear operator $\mathcal{L}_1$, with parameters $\mathbf{r}$ and $\boldsymbol{\alpha}$, acting on the variables $\mathbf{r}''$, $\boldsymbol{\alpha}''$ and $\lambda$ of the spectral radiance, such that $B_T(\mathbf{r},\boldsymbol{\alpha}) = \mathcal{L}_1\{L_T\}$.

## A.2. The point spread function of light pollution

The radiance of the night sky due to light pollution $L(\mathbf{r}'',\boldsymbol{\alpha}'', \lambda)$ is caused by the emissions of the artificial light sources located in the surrounding territory, up to distances that can reach hundreds of kilometres.

The emissions of any source can be characterized by its spectral radiance, $L_s(\mathbf{r}',\boldsymbol{\alpha}', \lambda)$, where $\mathbf{r}'$ is the source location and $\boldsymbol{\alpha}' = (z',\varphi')$ are the zenith angle and the azimuth, respectively, of the emission direction measured in the source's reference frame. The light emitted by the source propagates through the atmosphere being progressively attenuated by absorption and scattering. A fraction of this light, after undergoing one or several scattering events, propagates in the direction of the observer and contributes to the build-up of the light pollution radiance $L(\mathbf{r}'',\boldsymbol{\alpha}'', \lambda)$. The amount of scattered light is determined by the properties of the molecular and aerosol components of the atmosphere.

The radiance $L_s(\mathbf{r}',\boldsymbol{\alpha}', \lambda)$ of the emissions of the artificial lamps used in outdoor lighting is sufficiently low to ensure that the propagation of light through the atmosphere takes place in a linear regime, avoiding nonlinear phenomena like thermal lensing, two-photon absorption and similar. This means that the light pollution spectral radiance arriving at the observer can be expressed as the sum of the contributions of all surrounding sources by means of a weighted integral of the kind

$$L(\mathbf{r}'',\boldsymbol{\alpha}'', \lambda) = \int_{\Omega'} \int_{A'} G(\mathbf{r}'',\boldsymbol{\alpha}''; \mathbf{r}',\boldsymbol{\alpha}'; \lambda)\, L_s(\mathbf{r}',\boldsymbol{\alpha}', \lambda)\, \mathrm{d}^2\boldsymbol{\alpha}'\, \mathrm{d}^2\mathbf{r}, \tag{A 4}$$

where the kernel $G(\mathbf{r}'',\boldsymbol{\alpha}''; \mathbf{r}',\boldsymbol{\alpha}'; \lambda)$ is the spatial and angular PSF of the light pollution radiance, that is, the radiance produced at the observer location $\mathbf{r}''$ from the direction $\boldsymbol{\alpha}''$ due to a unit amplitude artificial light source located at $\mathbf{r}'$ emitting in the direction $\boldsymbol{\alpha}'$. The integrals are extended to all possible emission directions in the sources reference frame, $\Omega'$, and to the whole territory containing artificial lights, $A'$. The precise form of the function $G(\mathbf{r}'',\boldsymbol{\alpha}''; \mathbf{r}',\boldsymbol{\alpha}'; \lambda)$ depends on the details of the radiative transfer model used to compute the atmospheric propagation (e.g. single versus multiple scattering), the particular conditions of the atmosphere (specially the aerosol concentration profiles, albedos and angular phase functions), the spectral reflectance of the intervening terrain (through its bidirectional reflectance distribution function), and also on the presence of obstacles that could block the propagation of light in certain emission directions before it gets scattered towards the observer. A set of widely used models are available to determine $G(\mathbf{r}'',\boldsymbol{\alpha}''; \mathbf{r}',\boldsymbol{\alpha}'; \lambda)$ for different atmospheres and with different levels of analytic and numerical complexity. The interested reader may want to consult [23–31] for full details of the most used ones.

Equation (A 4) can be interpreted as the result of an integral linear operator, $\mathcal{L}_2$, with parameters $\mathbf{r}''$ and $\boldsymbol{\alpha}''$, acting on the variables $\mathbf{r}'$, $\boldsymbol{\alpha}'$ and $\lambda$ of the spectral radiance of the outdoor lights, such that $L(\mathbf{r}'',\boldsymbol{\alpha}'',\lambda) = \mathcal{L}_2\{L_s\}$. Combining this equation with the one for the light pollution indicator, $B(\mathbf{r},\boldsymbol{\alpha}) = \mathcal{L}_1\{L\}$, one immediately gets $B(\mathbf{r},\boldsymbol{\alpha}) = \mathcal{L}_1\{\mathcal{L}_2\{L_s\}\} = \mathcal{L}\{L_s\}$, where $\mathcal{L} = \mathcal{L}_1 \circ \mathcal{L}_2$ is the linear operator resulting from the composition of the actions of $\mathcal{L}_1$ and $\mathcal{L}_2$. The explicit form of this action is

$$B(\mathbf{r},\boldsymbol{\alpha}) = \int_\Lambda \int_A \int_\Omega \int_{\Omega'} \int_{A'} R(\mathbf{r},\boldsymbol{\alpha}; \mathbf{r}'',\boldsymbol{\alpha}''; \lambda) G(\mathbf{r}'',\boldsymbol{\alpha}''; \mathbf{r}',\boldsymbol{\alpha}'; \lambda)\, L_s(\mathbf{r}',\boldsymbol{\alpha}', \lambda)\, \mathrm{d}^2\boldsymbol{\alpha}'\, \mathrm{d}^2\mathbf{r}'\, \mathrm{d}^2\boldsymbol{\alpha}''\mathrm{d}^2\mathbf{r}''\mathrm{d}\lambda. \tag{A 5}$$

A simplified and useful expression can be obtained by making the only approximation we will use to apply this model, namely, that the artificial light sources are factorable [19] such that

$$L_s(\mathbf{r}',\boldsymbol{\alpha}', \lambda) = L_1(\mathbf{r}')\, L_2(\boldsymbol{\alpha}', \lambda). \tag{A 6}$$

This means that the light sources in the relevant surrounding territory have the same angular and spectral emission patterns, $L_2(\boldsymbol{\alpha}', \lambda)$, just differing in the absolute amount of emissions, $L_1(\mathbf{r}')$. Whereas this approximation may not hold for individual sources, it is expected to be valid for the overall emissions from patches of the territory with the typical pixel size of satellite imagery (of order of hundreds of metres long and wide). The dimensions of $L_1$ and $L_2$ can be arbitrarily chosen, as far as their product, $L_s$, has dimensions of spectral radiance.

By substituting equation (A 6) into equation (A 5) we get

$$B(\mathbf{r},\boldsymbol{\alpha}) = \int_\Lambda \int_A \int_\Omega \int_{\Omega'} \int_{A'} R(\mathbf{r},\boldsymbol{\alpha}; \mathbf{r}'',\boldsymbol{\alpha}''; \lambda) G(\mathbf{r}'',\boldsymbol{\alpha}''; \mathbf{r}',\boldsymbol{\alpha}'; \lambda)\, L_1(\mathbf{r}')\, L_2(\boldsymbol{\alpha}', \lambda)\, \mathrm{d}^2\boldsymbol{\alpha}'\, \mathrm{d}^2\mathbf{r}'\, \mathrm{d}^2\boldsymbol{\alpha}''\mathrm{d}^2\mathbf{r}''\mathrm{d}\lambda. \tag{A 7}$$

Changing the order of integration and regrouping terms

$$B(\mathbf{r},\boldsymbol{\alpha}) = \int_{A'} \left[ \iint_A \int_\Omega \int_{\Omega'} \int_\Lambda R(\mathbf{r},\boldsymbol{\alpha}; \mathbf{r}'',\boldsymbol{\alpha}''; \lambda) G(\mathbf{r}'',\boldsymbol{\alpha}''; \mathbf{r}',\boldsymbol{\alpha}'; \lambda)\, L_2(\boldsymbol{\alpha}', \lambda)\, \mathrm{d}^2\boldsymbol{\alpha}'\, \mathrm{d}^2\boldsymbol{\alpha}''\mathrm{d}^2\mathbf{r}''\mathrm{d}\lambda \right] L_1(\mathbf{r}')\, \mathrm{d}^2\mathbf{r}' \tag{A 8}$$

or, equivalently, and simplifying the notation using $B(\mathbf{r},\boldsymbol{\alpha}) \equiv B(\mathbf{r})$ (the value of $\boldsymbol{\alpha}$ can be considered as an implicit parameter of $B$ and $K$)

$$B(\mathbf{r}) = \int_{A'} K(\mathbf{r},\mathbf{r}')\, L_1(\mathbf{r}')\, \mathrm{d}^2\mathbf{r}, \tag{A 9}$$

where $K(\mathbf{r},\mathbf{r}')$ is the overall PSF

$$K(\mathbf{r},\mathbf{r}') = \int_A \int_\Omega \int_{\Omega'} \int_\Lambda R(\mathbf{r},\boldsymbol{\alpha}; \mathbf{r}'',\boldsymbol{\alpha}''; \lambda) G(\mathbf{r}'',\boldsymbol{\alpha}''; \mathbf{r}',\boldsymbol{\alpha}'; \lambda) \, L_2(\boldsymbol{\alpha}', \lambda) \, \mathrm{d}^2\boldsymbol{\alpha}' \, \mathrm{d}^2\boldsymbol{\alpha}'' \mathrm{d}^2\mathbf{r}'' \mathrm{d}\lambda, \qquad (A\,10)$$

This PSF depends on the kind of indicator and measuring instrument through $R(\mathbf{r},\boldsymbol{\alpha}; \mathbf{r}'', \boldsymbol{\alpha}''; \lambda)$, on the atmospheric and geographical conditions through $G(\mathbf{r}'', \boldsymbol{\alpha}''; \mathbf{r}', \boldsymbol{\alpha}'; \lambda)$, and on the luminaires angular and spectral radiation pattern through $L_2(\boldsymbol{\alpha}', \lambda)$. For any set of such conditions a specific PSF shall be calculated.

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
