## [Reviewer comments · Royal Society Open Science]

Review History

RSOS-201501.R0 (Original submission)

Review form: Reviewer 1

Is the manuscript scientifically sound in its present form?

Yes

Are the interpretations and conclusions justified by the results?

Yes

Is the language acceptable?

Yes

Do you have any ethical concerns with this paper?

No

Have you any concerns about statistical analyses in this paper?

No

Recommendation?

Accept with minor revision (please list in comments)

Comments to the Author(s)

Dear Authors,

I think the paper extremely needs it. The title might be a bit confusing on them of you expect a technical paper and what you find on the paper is more a technical explanation of why current laws do not work, than how to calculate the light pollution on an observatory. Therefore, I think you should change the title to something that reflects the true content of the paper. I suggest you for example I suggest you something like: Implications of the linear nature of the emissions on current and future regulations to protect the darkness of the night.

For me, the word "regulations", "implications" and "linear" are fundamental to reflect the content of the paper. If they are not included, the authors risk the paper will be only considered by a specialist. In that sense, on the conclusions at least, I would suggest the authors to explain in plain words understandable by policy makers the meaning of "linear", in other words the total amount of light emitted with respect to the distance to the source and maybe make some similarity with other fields of examples, like other kinds of pollution. This paper has to be didactic, and in that sense the conclusions is the place to explain plainly the real consequences of continuing with the current style of law. Another suggestion is maybe to consider the absurd consequences of the current laws. Like, according to them it would be legal to install hundreds of street lights just under an observatory if they have the legal ULOR and illumination levels.

Review form: Reviewer 2

Is the manuscript scientifically sound in its present form?

Yes

Are the interpretations and conclusions justified by the results?

Yes

Is the language acceptable?

Yes

Do you have any ethical concerns with this paper?

No

Have you any concerns about statistical analyses in this paper?

No

Recommendation?

Accept with minor revision (please list in comments)

Comments to the Author(s)

The paper is an important contribution and I highly recommend its publication. I especially congratulate the authors on the melding of science and social and political policy, which is sorely needed these days. The following suggestions for minor revision I believe make the paper more readable. It is not an easy read, in part because of the writing style which is a bit wordy, but definitely communicates adequately the main points. I admire the synthesis of previous work presented within a rigorous mathematical framework with an objective that is long overdue—moving from "do the best you can with new installations one light at a time" to a sky glow standard that, if exceeded, triggers a required mitigation response to existing outdoor lighting.

page 2 line 32, suggest replace "cultural enjoyment" with "preservation of cultural heritage".

page 2 line 36 "inmission" this is not an English word, more of a coined word, suggest replacing with "ambient artificial sky glow"

page 2 line 39 "take into account for other protection needs" this is not very informative in a summary, just list one or two, like "protect nocturnal biology"

page 3 lines 1-23 this is a rather lengthy introduction but considering the title of the paper is warranted

page 3 line 36 There is an important documented lighting ordinance that should be mentioned that actually has worked, in Flagstaff, Arizona USA a small area around US Naval Observatory and Anderson Mesa have a cap on lumens per acre depending on the distance from the observatory. See section 1.2 and Figure 1 in

<https://iopscience.iop.org/article/10.1086/597625/meta>

Also, Pima County, Arizona limits lumens but only on developed parcels, not total acres. See Section 401 in

https://webcms.pima.gov/UserFiles/Servers/Server_6/File/Government/Development%20Services/Building/OLC.pdf

Page 3 line 47 is modeling the only alternative to determining ambient sky glow over a region? What about monitoring? Later in the paper an analogy is drawn to air quality, the US Clean Air Act requires continuous ambient air monitoring,

Page 4 lines 1-30. The definition of terms and their relationship in the physical world is a little hard to follow. By referring to Figure 1 in reference [20] I understood but perhaps a diagram would assist the reader in keeping track of the variables, or refer the reader to reference [20] here rather than later.

page 6 line 14 the use of VIIRS DNB composites. A word of caution should probably be inserted here: VIIRS DNB data are not directly correlated with actual upward radiance from outdoor lighting sources. This is because 1) DNB does not extend to wavelengths shorter than 500 nm and is sensitive to near IR, and 2) upward radiance reaching the satellite is subject to atmospheric and ground albedo variability so month to month variability in the composites may not be entirely due to variability in the sources. See <https://www.mdpi.com/2072-4292/10/12/1964> Yearly composites may be somewhat more stable, but this is still worth mentioning.

page 8 lines 6-8 Besides observatories, an entire region could be zoned for establishing admissible limits. For the U.S. Clean Air Act, they are called standards. For daytime visibility areas are classified as to sensitivity. Class I areas are the most sensitive and have the strictest standards. Protected areas such as wilderness, biological preserves, and parks might be zoned for the strictest standard. The IDA Model Lighting Ordinance outlines zones with different standards for lighting installations, but stopped short of establishing standards for indicators of light pollution. These existing examples may be worth mentioning. see

<https://www.epa.gov/visibility/regional-haze-program>

<https://www.darksky.org/our-work/lighting/public-policy/mlo/>

page 8 lines 16-23. here the authors are moving toward opinion rather than science. I think that bringing up these questions is appropriate, as it makes the reader think. However, I would delete the last phrase "probably richer and better equipped" to keep the paper apolitical.

Decision letter (RSOS-201501.R0)

Dear Dr Bará

On behalf of the Editors, we are pleased to inform you that your Manuscript RSOS-201501 "Protecting the night sky darkness in astronomical observatories: a linear systems approach" has been accepted for publication in Royal Society Open Science subject to minor revision in accordance with the referees' reports. Please find the referees' comments along with any feedback from the Editors below my signature.

Please submit your revised manuscript and required files (see below) no later than 7 days from today's (ie 04-Nov-2020) date. Note: the ScholarOne system will 'lock' if submission of the revision is attempted 7 or more days after the deadline. If you do not think you will be able to meet this deadline please contact the editorial office immediately.

on behalf of Professor Mark McCaughrean (Associate Editor) and Rob Ivison (Subject Editor)
openscience@royalsociety.org

Associate Editor Comments to Author (Professor Mark McCaughrean):

Comments to the Author:

On the basis of the reviews conducted by the referees and assuming that their minor comments are taken into account during revision, I'm happy for this paper to be accepted for publication – it clearly makes an important contribution to the issue of light pollution in terms of modelling and regulation.

Reviewer comments to Author:

Reviewer: 1

Comments to the Author(s)

Dear Authors,

I think the paper extremely needs it. The title might be a bit confusing on term of you expect a technical paper and what you find on the paper is more a technical explanation of why current laws do not work, than how to calculate the light pollution on a observatory. Therefore, I think you should change the title to some thing the reflect the true content of the paper. I suggest you for

example I suggest you something like: Implications of the linear nature of the emissions on current and future regulations to protect the darkness of the night.

For me, the word "regulations", "implications" and "linear" are fundamental to reflect the content of the paper. If they are not included, the authors risk the paper will be only consider by specialist. In that sens, on the conclusions at leas, I would suggest the authors to explain in plain words understandable by police makers the meaning of "linear", in other words the total amount of light emited withed by the distance to the source and maybe make some simil with other fields of examples, like other kinds of pollution. This paper has to be didactic, and in that snens the conclusions is the place to explain plainly the real consequences of continuing with the current style of law. Another suggestion is maybe to consider the absurdness consequences of the current laws. Like, according with them would be legal to install hundreds of street lights just under an observatory if they have the legal ULOR and illumination levels.

Reviewer: 2

Comments to the Author(s)

The paper is an important contribution and I highly recommend its publication. I especially congratulate the authors on the melding of science and social and political policy, which is sorely needed these days. The following suggestions for minor revision I believe make the paper more readable. It is not an easy read, in part because of the writing style which is a bit wordy, but definitely communicates adequately the main points. I admire the synthesis of previous work presented within a rigorous mathematical framework with an objective that is long overdue--moving from "do the best you can with new installations one light at a time" to a sky glow standard that, if exceeded, triggers a required mitigation response to existing outdoor lighting.

page 2 line 32, suggest replace "cultural enjoyment" with "preservation of cultural heritage".

page 2 line 36 "inmission" this is not an English word, more of a coined word, suggest replacing with "ambient artificial sky glow"

page 2 line 39 "take into account for other protection needs" this is not very informative in a summary, just list one or two, like "protect nocturnal biology"

page 3 lines 1-23 this is a rather lengthy introduction but considering the title of the paper is warranted

page 3 line 36 There is an important documented lighting ordinance that should be mentioned that actually has worked, in Flagstaff, Arizona USA a small area around US Naval Observatory and Anderson Mesa have a cap on lumens per acre depending on the distance from the observatory. See section 1.2 and Figure 1 in

<https://iopscience.iop.org/article/10.1086/597625/meta>

Also, Pima County, Arizona limits lumens but only on developed parcels, not total acres. See Section 401 in

https://webcms.pima.gov/UserFiles/Servers/Server_6/File/Government/Development%20Services/Building/OLC.pdf

Page 3 line 47 is modeling the only alternative to determining ambient sky glow over a region? What about monitoring? Later in the paper an analogy is drawn to air quality, the US Clean Air Act requires continuous ambient air monitoring,

Page 4 lines 1-30. The definition of terms and their relationship in the physical world is a little hard to follow. By referring to Figure 1 in reference [20] I understood but perhaps a diagram would assist the reader in keeping track of the variables, or refer the reader to reference [20] here rather than later.

page 6 line 14 the use of VIIRS DNB composites. A word of caution should probably be inserted here: VIIRS DNB data are not directly correlated with actual upward radiance from outdoor lighting sources. This is because 1) DNB does not extend to wavelengths shorter than 500 nm and

is sensitive to near IR, and 2) upward radiance reaching the satellite is subject to atmospheric and ground albedo variability so month to month variability in the composites may not be entirely due to variability in the sources. See <https://www.mdpi.com/2072-4292/10/12/1964> Yearly composites may be somewhat more stable, but this is still worth mentioning.

page 8 lines 6-8 Besides observatories, an entire region could be zoned for establishing admissible limits. For the U.S. Clean Air Act, they are called standards. For daytime visibility areas are classified as to sensitivity. Class I areas are the most sensitive and have the strictest standards. Protected areas such as wilderness, biological preserves, and parks might be zoned for the strictest standard. The IDA Model Lighting Ordinance outlines zones with different standards for lighting installations, but stopped short of establishing standards for indicators of light pollution. These existing examples may be worth mentioning. see

<https://www.epa.gov/visibility/regional-haze-program>

<https://www.darksky.org/our-work/lighting/public-policy/mlo/>

page 8 lines 16-23. here the authors are moving toward opinion rather than science. I think that bringing up these questions is appropriate, as it makes the reader think. However, I would delete the last phrase "probably richer and better equipped" to keep the paper apolitical.

===PREPARING YOUR MANUSCRIPT===

===PREPARING YOUR REVISION IN SCHOLARONE===

Author's Response to Decision Letter for (RSOS-201501.R0)

See Appendix A.

Decision letter (RSOS-201501.R1)

Dear Dr Bará,

It is a pleasure to accept your manuscript entitled "A linear systems approach to protect the night sky: implications for current and future regulations" in its current form for publication in Royal Society Open Science. The comments of the reviewer(s) who reviewed your manuscript are included at the foot of this letter.

Bear in mind that the journal provides a 'light touch' copy-editing service, though it has been observed that you may benefit from giving your proof a close read to ensure clarity/readability.

on behalf of Professor Mark McCaughrean (Associate Editor) and Rob Ivison (Subject Editor)
openscience@royalsociety.org

Associate Editor Comments to Author (Professor Mark McCaughrean):
Associate Editor
Comments to the Author:

The editors are now happy to accept this paper without further revision, as the remarks and suggestions made by the reviewers have been carefully considered and folded into the paper accordingly.

Appendix A

MS Reference Number: RSOS-201501 (Revised version)

MS Title: Protecting the night sky darkness in astronomical observatories: a linear systems approach

MS Authors: Falchi, Fabio; Bará, Salvador

Dear Editor,

We acknowledge the encouraging and useful comments of Assoc. Editor Prof. Mark McCaughrean and both Reviewers, and we have modified our manuscript according to them. Please find below a detailed account of the changes made in the revised version.

Best regards,

Salva Bará and Fabio Falchi

Associate Editor Comments to Author (Professor Mark McCaughrean):

Comments to the Author:

> On the basis of the reviews conducted by the referees and assuming that their minor comments are taken into account during revision, I'm happy for this paper to be accepted for publication – it clearly makes an important contribution to the issue of light pollution in terms of modelling and regulation.

Answer: We have revised the paper taking into account all Reviewers' comments. A detailed list of the changes made in the revised version can be found below. These changes are highlighted in red characters in the main text. We also acknowledge the comment of Reviewer 2 about the readability of this paper, and we have tried to improve it in the revised version with minor wording changes that do not modify the content of this work. These style revisions are highlighted in blue.

Reviewer comments to Author:

Reviewer: 1

Comments to the Author(s)

> Dear Authors,

I think the paper extremely needs it. The title might be a bit confusing on therm of you expect a technical paper and what you find on the paper is more a technical explanation of why current laws do not work, than how to calculate the light pollution on a observatory. Therefore,

I think you should change the title to some thing the reflect the basic content of the paper. I suggest you for example I suggest you something like: Implications of the linear nature of the emissions on current and future regulations to protect the darkness of the night. For me, the word "regulations", "implications" and "linear" are fundamental to reflect the content of the paper. If they are not included, the authors risk the paper will be only consider by specialist.

Answer: We acknowledge the convenience of adopting a title as much informative as possible, and gladly accept the Reviewer's criticism. We have modified the original title to: "A linear systems approach to protect the night sky: implications for current and future regulations".

> In that sens, on the conclusions at leas, I would suggest the authors to explain in plain words understandable by police makers the meaning of "linear", in other words the total amount of light emited withed by the distance to the source and maybe make some simil with other fields of examples, like other kinds of pollution. This paper has to be didactic, and in that snens the conclusions is the place to explain plainly the real consequences of continuing with the current style of law.

Another suggestion is maybe to consider the absurdness consequences of the current laws. Like, according with them would be legal to install hundreds of street lights just under an observatory if they have the legal ULOR and illumination levels.

Answer: We agree, and we have included a specific statement in the conclusions of the revised version.

Reviewer: 2

Comments to the Author(s)

> The paper is an important contribution and I highly recommend its publication. I especially congratulate the authors on the melding of science and social and political policy, which is sorely needed these days. The following suggestions for minor revision I believe make the paper more readable. It is not an easy read, in part because of the writing style which is a bit wordy, but definitely communicates adequately the main points. I admire the synthesis of previous work presented within a rigorous mathematical framework with an objective that is long overdue--moving from "do the best you can with new installations one light at a time" to a sky glow standard that, if exceeded, triggers a required mitigation response to existing outdoor lighting.

Answer: We really appreciate this encouraging assessment of our work. We also thank the Reviewer for pointing out the difficulties that our writing style may create for an easy read of this paper. We have tried to improve somewhat the wording in this revised version. The

changes of style that do not modify the content of the paper are highlighted in blue. We hope this can alleviate, at least partially, the readability problem. The changes directly related to the Reviewers' comments are highlighted in red.

> page 2 line 32, suggest replace "cultural enjoyment" with "preservation of cultural heritage".

Answer: Done.

> page 2 line 36 "inmission" this is not an English word, more of a coined word, suggest replacing with "ambient artificial sky glow"

Answer: Done. We regret having misspelled "immissions". Although an obsolete word according to classical English dictionaries, it is being commonly used in modern environmental literature and some European regulations (with adapted forms as in the German law *Bundes-Immissionsschutzgesetz* (BImSchG) or the Spanish term *inmisiones medioambientales*. The general meaning of this term is well described in e.g. this page of a private firm of environmental services:

<https://enviraiot.com/immission-vs-emission-what-are-their-differences/>

> page 2 line 39 "take into account for other protection needs" this is not very informative in a summary, just list one or two, like "protect nocturnal biology"

Answer: Done.

> page 3 lines 1-23 this is a rather lengthy introduction but considering the title of the paper is warranted

Answer: We agree it is a bit longer than a typical introduction would require. However, we wanted to put in context our work, paying special attention to the readers that might be aware of the general problem of light pollution but not be acquainted with the pressing problems faced by some optical astronomical observatories. We have tried to make some changes of style to improve readability.

> page 3 line 36 *There is an important documented lighting ordinance that should be mentioned that actually has worked, in Flagstaff, Arizona USA a small area around US Naval Observatory and Anderson Mesa have a cap on lumens per acre depending on the distance from the observatory. See section 1.2 and Figure 1 in*

<https://iopscience.iop.org/article/10.1086/597625/meta>

Also, Pima County, Arizona limits lumens but only on developed parcels, not total acres. See Section 401 in

https://webcms.pima.gov/UserFiles/Servers/Server_6/File/Government/Development%20Services/Building/OLC.pdf

Answer: We thank the Reviewer for calling our attention on these examples of territorial lumen caps that we overlooked in our literature search. We have included an explicit mention to them in that paragraph and added the corresponding references.

> Page 3 line 47 is modeling the only alternative to determining ambient sky glow over a region? What about monitoring? Later in the paper an analogy is drawn to air quality, the US Clean Air Act requires continuous ambient air monitoring

Answer: The quantitative model referred to in this paragraph aims to determine the percent contribution of every patch of the territory to the overall levels of sky brightness over the protected site. This model is instrumental for long-term planning of emissions contention or reduction, allocating the corresponding quotas among the intervening stakeholders. Permanent, direct monitoring of the evolution of the night sky brightness is of course required to check compliance: we assumed it implicitly, but we make it now explicit in the revised version. We have also included a small paragraph at the end of Section 2 explaining more clearly the scope of this paper, and the contents dealt with in each section.

> Page 4 lines 1-30. The definition of terms and their relationship in the physical world is a little hard to follow. By referring to Figure 1 in reference [20] I understood but perhaps a diagram would assist the reader in keeping track of the variables, or refer the reader to reference [20] here rather than later.

Answer: We agree that quoting at the beginning of this section references [19,20], could be useful for readers. In the revised version we mention them in the first line of the second paragraph.

> page 6 line 14 the use of VIIRS DNB composites. A word of caution should probably be inserted here: VIIRS DNB data are not directly correlated with actual upward radiance from outdoor lighting sources. This is because 1) DNB does not extend to wavelengths shorter than 500 nm and is sensitive to near IR, and 2) upward radiance reaching the satellite is subject to atmospheric and ground albedo variability so month to month variability in the composites may not be entirely due to variability in the sources. See

<https://www.mdpi.com/2072-4292/10/12/1964> Yearly composites may be somewhat more stable, but this is still worth mentioning.

Answer: We agree that caution should be exercised when using VIIRS-DNB datasets, due to the reasons mentioned by the Reviewer. We have included some comments on this at the end of Section 3, not to interrupt the discourse in former p. 6, line 14, which dealt with geometric aspects of map reprojections from WGS to UTM coordinate reference systems. We have added the reference suggested by the Reviewer. Note however that the *relative* contributions of municipalities or districts to the light pollution over the observatories calculated following the approach described in this paper are relatively robust against (1), if the lighting technologies mix used in every pixel of the territory can be approximately considered constant. Regarding to (2), we don't consider it as a drawback: as a matter of fact, the emissions that are relevant for light pollution are not just the raw emissions of the luminaires, but the effective amount of light radiated from each pixel of the territory. In this sense, the fact that VIIRS-DNB shows a variability that can be taken into account in planning is an asset rather than something to correct.

> page 8 lines 6-8 Besides observatories, an entire region could be zoned for establishing admissible limits. For the U.S. Clean Air Act, they are called standards. For daytime visibility areas are classified as to sensitivity. Class I areas are the most sensitive and have the strictest standards. Protected areas such as wilderness, biological preserves, and parks might be zoned for the strictest standard. The IDA Model Lighting Ordinance outlines zones with different standards for lighting installations, but stopped short of establishing standards for indicators of light pollution. These existing examples may be worth mentioning. see <https://www.epa.gov/visibility/regional-haze-program> <https://www.darksky.org/our-work/lighting/public-policy/mlo/>

Answer: We fully agree. As a matter of fact, the general formalism for linking these territorial indicators to the source emissions has been presented in reference [19]. We have included a comment on this at the end of the second paragraph of Section 4.

> page 8 lines 16-23. here the authors are moving toward opinion rather than science. I think that bringing up these questions is appropriate, as it makes the reader think. However, I would delete the last phrase "probably richer and better equipped" to keep the paper apolitical.

Answer: In these lines we intended to call the attention of the readers to the fact that the decisions regarding public lighting cannot be deterministically deduced from a map of relative contributions of the municipalities and districts alone, because other relevant

criteria have to be taken into account. We see no problem in deleting the phrase suggested by the Reviewer, and so we did. That sentence was written thinking in small communities nearby first-class astronomical observatories located in relatively unpopulated areas. The (necessary and urgent) discussion of the social justice issues posed by site preservation, be them astronomical observatories or natural parks, is beyond the scope of this paper.

OTHER MINOR CHANGES

- A funding source was added.
- Figure 2 was slightly zoomed for the readers' convenience and the position of the observing site was added.